# Prenatal Diagnosis Using Chromosomal Microarray Analysis in High-Risk Pregnancies

**DOI:** 10.3390/jcm11133624

**Published:** 2022-06-23

**Authors:** Ching-Hua Hsiao, Jia-Shing Chen, Yu-Ming Shiao, Yann-Jang Chen, Ching-Hsuan Chen, Woei-Chyn Chu, Yi-Cheng Wu

**Affiliations:** 1Department of Biomedical Engineering, National Yang Ming Chiao Tung University, Taipei 112, Taiwan; wchu@nycu.edu.tw (W.-C.C.); wu102007@gmail.com (Y.-C.W.); 2Department of Obstetrics and Gynecology, Taipei City Hospital, Women and Children Campus, Taipei 100, Taiwan; DAZ07@tpech.gov.tw; 3School of Medicine for International Students, I-Shou University, Kaohsiung 840, Taiwan; cjshing77@gmail.com; 4Department of Bioscience Technology, Chung Yuan Christian University, Taoyuan 320, Taiwan; yumingshiao@gmail.com; 5Union Clinical Laboratory, Taipei 106, Taiwan; 6Department of Life Sciences and Institute of Genome Sciences, National Yang Ming Chiao Tung University, Taipei 112, Taiwan; yjchen0206@nycu.edu.tw; 7 Department of Obstetrics and Gynecology, Ultrasound Center of Taiwan IVF Group, Ton-Yen General Hospital, Zhubei 302, Taiwan

**Keywords:** chromosomal microarray analysis (CMA), copy number variants (CNVs), variants of unknown significance (VOUS), amniotic fluid (AF), chorionic villus sampling (CVS)

## Abstract

***Background***: To assess the value of chromosomal microarray analysis (CMA) during the prenatal diagnosis of high-risk pregnancies. ***Methods***: Between January 2016 and November 2021, we included 178 chorionic villi and 859 amniocentesis samples from consecutive cases at a multiple tertiary hospital. Each of these high-risk singleton pregnancies had at least one of the following indications: (1) advanced maternal age (AMA; ≥35 years; 546, 52.7%); (2) fetal structural abnormality on ultrasound (197, 19.0%); (3) high-risk first- or second-trimester Down syndrome screen (189, 18.2%), including increased nuchal translucency (≥3.5 mm; 90, 8.7%); or (4) previous pregnancy, child, or family history (105, 10.1%) affected by chromosomal abnormality or genetic disorder. Both G-banding karyotype analysis and CMA were performed. DNA was extracted directly and examined with oligonucleotide array-based comparative genomic hybridization. ***Results***: Aneuploidies were detected by both G-banding karyotyping and CMA in 42/1037 (4.05%) cases. Among the 979 cases with normal karyotypes, 110 (10.6%) cases had copy number variants (CNVs) in CMA, including 30 (2.9%) cases with reported pathogenic and likely pathogenic CNVs ≥ 400 kb, 37 (3.6%) with nonreported VOUS, benign, or likely benign CNVs ≥ 400 kb, and 43 (4.1%) with nonreported CNVs < 400 kb. Of the 58 (5.6%) cases with aneuploidy rearrangements, 42 (4.1%) were diagnosed by both G-banding karyotyping and CMA; four inversions, six balanced translocations, and six low mosaic rates were not detected with CMA. ***Conclusions***: CMA is an effective first step for the prenatal diagnosis of high-risk pregnancies with fetal structural anomalies found in ultrasonography or upon positive findings.

## 1. Background

Prenatal cytogenetic testing is currently offered most often to couples at high risk of having a child with a chromosomal abnormality. Cytogenetic fetal karyotyping can detect aneuploidy and large chromosomal rearrangements of up to 5–10 megabases (Mb), and was previously the “gold standard” prenatal genetic testing. Chromosomal microarray analysis (CMA) is a cytogenetic molecular technique that has a high detection rate for microscopic and submicroscopic chromosomal aberrations in patients with neurodevelopmental disorders (10–14%) [1]. CMA is used to assess microdeletions and microduplications called copy number variants (CNVs), which can be as small as 50–100 kb. Thus, CMA provides a ~100-fold higher resolution than conventional karyotyping, depending on the probe spacing and CMA platform used. The ability to examine the genome at this high resolution has resulted in the discovery of widespread CNVs in the human genome, including both polymorphic variations in healthy individuals and novel pathogenic copy number imbalances [2], and this has had a major impact on genetic diagnosis over the past decade [3]. Furthermore, CMA also provides additional clinically useful information in approximately 5% (range: 2.3–8.3%) of cases [4,5].

CMA enables the detection of smaller pathogenic chromosomal variants that are undetectable using standard cytogenetic analyses, and it can be highly customized and is amenable to high throughput. However, a potential drawback of CMA is that it does not allow for the detection of balanced chromosomal rearrangements, triploidy, and some instances of mosaicism. The biggest challenge that limits CMA is the detection of chromosomal variants of uncertain significance (VOUS). The difference in the reported prevalence of only pathogenic CNVs and the prevalence of both VOUS and pathogenic CNVs is likely to increase with the use of high-resolution, genome-wide array platforms, which have been applied more often in recent studies. However, unlike single nucleotide polymorphism (SNP) arrays, oligo-based CMA cannot identify balanced translocations, inversions, uniparental disomy, or polyploidy [3].

A pregnancy can be suspected of having a high risk of a chromosomal abnormality after screening due to a family history of chromosomal abnormalities or the detection of a structural anomaly in a prenatal ultrasound. CMA has now become the first-tier technique for the genetic follow-up of fetal structural anomalies identified by ultrasonography [6]. Recent studies have reported pathogenic CNVs at a rate of 0–15.0% among fetuses with increased nuchal translucency (NT; ≥2.5 to 3.5 mm, corresponding to the 95th to 99th percentile in the general population) or cystic hygroma [7,8]. The incidence of chromosomal anomalies in prenatal examinations is reported to be as high as 18–22% for all cases of congenital heart disease (CHD); the most common variants are trisomy 21 and 18, along with 22q11 microdeletion [9,10].

In this study, we report a multicenter, prospective cohort study of 1037 high-risk pregnancies that underwent prenatal diagnosis using G-banding karyotyping along with CMA. These analyses were carried out for suspected high-risk pregnancies or after observing a structural anomaly in an ultrasound during prenatal diagnosis.

## 2. Materials and Methods

### 2.1. Patients and Indications

We prospectively recruited 1037 high-risk singleton pregnancies after invasive diagnostic testing between January 2016 and December 2021. The inclusion criteria were high-risk singleton pregnancies with at least one of the following indications: (1) advanced maternal age (AMA; ≥35 years; 546, 52.7%); (2) fetal structural abnormality detected by ultrasound (197, 19.0%); (3) high risk on first- or second-trimester Down syndrome screening (189, 18.2%), including only increased nuchal translucency (≥3.5 mm; 90, 8.7%); or (4) previous pregnancy or live birth affected by or a family history of a chromosomal abnormality or another genetic disorder (105, 10.1%).

Both karyotype analysis and CMA were performed for all patients. The cohort comprised identified results from 1037 consecutive prenatal specimens; 929 cases underwent amniotic fluid (AF) sampling and 108 cases underwent chorionic villus sampling (CVS). AF studies were performed using direct (uncultured) specimens for 921 cases and cultured specimens for 8 cases. CVS studies were performed using uncultured specimens for 105 cases and cultured specimens for three cases. The subjects provided informed written consent for CMA when a positive indication was being considered. The Institutional Review Board approved this study (TCHIRB-10602101).

### 2.2. Prenatal Samples

Oral pretest counseling and written information was provided by a specialist in fetal medicine to inform the participants about the chance and implications of detecting aberrations with unknown or uncertain clinical relevance (variants of uncertain significance: VOUS) and incidental findings. Cases with high-risk indications underwent prenatal diagnosis using G-banding karyotyping along with CMA. If CMA analysis revealed an aberration, the parents’ blood samples were collected and analyzed to determine whether the fetal CNVs were inherited or de novo. CNVs of clinical significance or unknown significance were confirmed with fluorescence in situ hybridization (FISH) or quantitative real-time PCR (q-PCR), if necessary.

Maternal cell contamination is a possibility when studying uncultured amniotic fluid or CVS cells. Both the amniocytes and villi were dissected under a microscope to remove the maternal cells and decidua prior to DNA extraction.

### 2.3. DNA Preparation and Chromosomal Microarray Analysis

Chorionic villus sampling (CVS) and amniocentesis were performed for G-banding karyotype analysis and CMA. The material was immediately extracted from the uncultured villi (10–20 mg) and amniocentesis (5–10 mL) samples using Norgen’s Blood Genomic DNA Isolation Mini Kit (Norgen Biotek Corp., Thorold, ON, Canada).

For CMA, all samples were tested on targeted arrays using the CytoOneArray^®^ platform (versions 1.0 to 4.0) depending on the date of specimen receipt, as the design of these arrays changed over time (Phalanx Biotech Group Inc., Hsinchu, Taiwan) [11]. The targeted design of CytoOneArray^®^ can analyze more than 500 DNA regions associated with known genetic disorders, especially in Asian populations (https://www.phalanx.com.tw/reproductive-medicine/prenatal-testing/cyto-390/?lang=en (accessed on 1 March 2022)). Sample and reference genomic DNA (50 ng) were labeled with Cy5 (reference) or Cy3 (specimen) using a low-input DNA amplification and labeling kit. All experimental procedures were performed according to the manufacturer’s protocol. Scanning and image acquisition were carried out using an Agilent microarray scanner (G2565C/G2600D), and microarray image files were quantified using GenePix Pro 6 (Axon Instruments, Union City, CA, USA). Data analysis was performed using R version 3.4.1 (R Foundation for Statistical Computing, Vienna, Austria) and human genome version GRCh37 (hg19).

### 2.4. Interpretation and Reporting of CNVs

The detected CNVs were classified as (1) pathogenic, (2) likely pathogenic, (3) of uncertain significance, (4) likely benign, or (5) benign, along with standard definitions for each term, in accordance with the guidelines from the American College of Medical Genetics (ACMG) and the International Standard Cytogenomic Array (ISCA). Copy number gains or losses detected with CMA were evaluated systematically for clinical significance via comparison with public CNV databases and with our inhouse database. If aberrant CNVs were detected in fetal DNA, the parental DNA was further analyzed to determine the inheritance of variants in the family to provide information about the significance of VOUS. For clinical arrays, CNV gains or losses of at least ≥400 kb are reported, and CNVs smaller than these size limits for clinical significance are not reported [1,2,6,12,13].

The findings were interpreted according to whether the CNVs were described as pathogenic or benign in the scientific literature, general genome databases, integrated databases, and curated databases. Genomic structural variations were annotated using ClinVar (https://www.ncbi.nlm.nih.gov/clinvar/ (accessed on 1 March 2022)) and ClinGen (https://clinicalgenome.org/ (accessed on 1 March 2022)). Disease-causing genes, their functions, and their inheritance patterns were confirmed using the Online Mendelian Inheritance in Man (OMIM, http://www.omim.org/ (accessed on 1 March 2022)) Database of Genomic Variants (DGV http://dgt.tcag.ca/sgv/app/home (accessed on 1 March 2022)). The Database of Chromosomal Imbalance and Phenotype in Humans using Ensemble Resources (DECIPHER, https://decipher.sanger.ac.uk/ (accessed on 1 March 2022)) was used as a reference for known microdeletion and microduplication syndromes. In addition, the SFARI GENE database (https://gene.sfari.org/autdb/Welcome.do (accessed on 1 March 2022)) was employed to provide genetic information on evaluated CNVs that are associated with autism spectrum disorders (ASD), developmental delay (DD), and intellectual disability (ID) [2]. Additionally, a subgroup of the pathogenic CNVs was labeled as susceptibility variants, according to Rosenfeld et al. [14].

## 3. Results

The total number of cases enrolled with indications for invasive prenatal testing by CMA and conventional cytogenetics are presented in Figure 1. For all cases with findings of pathologic/unknown penetrance or VOUS, parental a-CGH analysis was performed to determine whether the findings were inherited or de novo. Furthermore, all fetuses with duplication or deletion findings were evaluated with detailed ultrasound scans to confirm the structural information related to the chromosomal abnormalities. Both the G-banding karyotyping and CMA results were normal for 869 of the 1037 cases. We excluded 43 cases with a normal karyotype and CNVs < 400 kb. Of the 110 cases with a normal karyotype and CNVs ≥ 400 kb, pathogenic variants and likely pathogenic variants were reported for 30 cases, and 37 cases had nonreported VOUS, benign, or likely benign variants. G-banding karyotype rearrangements were detected in 58 cases (58/1037, 5.6%), including 42 (4.1%) cases of aneuploidy, of which 14, 10, 3, 5, 3, and 2 cases were trisomy 21, 18, 13, monosomy X, 47,XXY, and 47,XXX, respectively. G-banding indicated six balanced-translocations, four inversions, and six mosaic aberrant karyotypes among cases with normal CMA results. Therefore, in our cohort, CMA produced an additional diagnostic yield of 2.9% (30/1037) for CNVs ≥ 400 kb, including reported pathogenic or likely pathogenic variants, 3.6% (37/1037) for nonreported variants, including VOUS, benign, and likely benign variants, and 4.1% (43/1037) for CNVs < 400 kb, corresponding to a total additional diagnostic yield of 10.6% (110/1037).

As shown in Table 1, among the 546 cases (52.7%; 546/1037) cases with AMA only (the major indication for prenatal testing in this cohort), 4% (22/546) of them showed chromosomal abnormality with CMA. However, only 1.1% (6/546) showed pathogenic/likely pathogenic CNVs. This indicated that AMA only is less relevant than other indications in Table 1. Among the 197 (19.0%, 197/1037) cases with abnormal ultrasound findings, reported CNVs ≥ 400 kb were detected in 11 cases (1.06%, 11/1037). The major anomalies observed on ultrasound included hydrocephalus, congenital fetal heart anomalies, megacystis, pyelectasis, gastroschisis, omphalocele, and the pentalogy of Cantrell. Reported CNVs ≥ 400 kb were detected in 13 (1.25%, 13/1037) of the 189 cases (18.2%, 189/1037) with high-risk Down syndrome screening results (≥1:270). All cases of trisomy 21, 18, and 13 diagnosed by G-banding and CMA were classified as high risk in Down syndrome screening (≥1:270). Thirteen of thirty (43.3%) cases with reported CNVs ≥ 400 kb had high-risk Down syndrome screening results (≥1:270). Reported and nonreported CNVs ≥ 400 kb were detected for one and five cases among the 10.1% (105/1037) of pregnancies with a family history or previous pregnancy with genetic abnormalities, respectively.

As shown in Table 2, pathogenic and likely pathogenic CNVs ≥ 400 kb were reported for 30 cases. Most of these cases presented multiple high-risk indications during chromosomal screening (NT ≥ 3.5 mm, 9 cases) or ultrasound scan anomalies (10 cases); 25 variants were de novo and five cases were pathogenic penetrance inheritance. The outcomes for these 30 cases were 20 terminations of pregnancy (TOP), 7 healthy births, 2 live births with signs of mental retardation, and 1 intrauterine fetal death at 32 weeks of gestation.

## 4. Discussion

Due to the incrementally higher diagnostic yield of CMA for chromosomal abnormalities compared to traditional G-banding karyotyping, numerous studies have used CMA to detect microdeletions and microduplications in high-risk pregnancies. Prenatal CMA is now more commonly employed in cases where fetal anomalies have been detected by ultrasound [15]. For isolated fetal anomalies, prenatal CMA has been shown to provide additional diagnostic yield over conventional karyotyping, and has thus been recommended in such situations. Metanalyses of CMA results in cases with isolated ultrasound defects indicate a pathogenic variant rate of ~5% [16]. Prenatal CMA identified clinically significant genomic alterations in 9.1% of cases with one or more abnormal ultrasound finding, and the majority of these variants were below the resolution of karyotyping, with the greatest yield observed for cardiac and renal anomalies [17,18]. Thus, it is likely that CMA will replace karyotyping in high-risk pregnancies [15]. Based on the available evidence, the American College of Obstetricians and Gynecologists and the Society of Maternal–Fetal Medicine recommend CMA should be performed instead of karyotyping in pregnancies with anomalous fetuses undergoing invasive testing [4,19].

Most clinical laboratories performing aCGH in postnatal studies report clinically significant imbalances in the range of 50–100 kb. The reporting size range is usually larger in prenatal studies, and it may vary according to the indication for testing [3]. A number of guidelines or recommendations have been developed in several countries. The practice guidelines of the Canadian College of Medical Geneticists–Society of Obstetricians and Gynecologists of Canada (CCMG-SOGC) reported that CNVs overlap completely with an established dosage-sensitive region. The Royal College of Pathologists, the British Society for Genetic Medicine, and the Royal College of Obstetricians and Gynecologists (RCP-BSGM-RCOG) also recommend the use of CMA in high-risk pregnancies [20,21]. VOUS above the size of the CMA cutoffs are only reported if there is significant supporting evidence that deletion or duplication of the region may be pathogenic. Secondary findings associated with a medically actionable disorder with childhood onset must be reported, whereas variants associated with adult-onset conditions are not reported unless requested by the parents or if disclosure could prevent serious harm to family members. In this study, the pathogenic dosage-sensitive region cutoffs used for the reported CNVs are in agreement with both the Canadian and British practice guidelines.

### 4.1. Overview of CMA Platforms and Interpretation of Prenatal Examinations

Clinical arrays are typically designed to detect imbalances of 20–50 kb in targeted regions (e.g., within known Mendelian genes) and imbalances of 100–250 kb in nontargeted (backbone) regions of the genome [1]. This study used the CytoOneArray^®^ cytogenomic microarray. This CMA (~33,000 probes of 60 nt) is based on a minimum consecutive probe number of 15 in pathogenic/likely pathogenic CNV hotspots, rather than spreading the probes uniformly within pathogenic/likely pathogenic gene regions. The target regions are selected based on pathogenic/likely pathogenic CNV frequency data in CNV databases to implement CMA designs with added targeted coverage of known disease-associated genes and regions (e.g., OMIM morbid genes). The nontargeted regions are designed as five consecutive probes at a distance of 10–30 kb per 1 Mb [22]. Prenatal array platforms conforming to the European and international consensus have also been established for a lower limit threshold of 400 kb across the genome [21]. The literature indicates that the platform, size filter cutoffs, and target regions of cytogenomic microarrays affect the detection of CNVs in prenatal diagnosis. The findings of this study underscore the significant benefits of the CMA platform, as well as the recommended size filter cutoffs and target regions used in data analysis, for the detection of CNVs in a cohort of prenatal cases. An overview of the recent literature on CMA platforms and the interpretation of prenatal examinations is shown in Table 3. We used the ACMG standards and guidelines for the interpretation and reporting of prenatal constitutional CNVs. Parental studies provide some of the most useful evidence on the clinical significance of the observed CNVs. The interpretation of a VOUS can be aided by information about whether the variant was inherited from a healthy parent or occurred de novo in the proband. Population studies suggest that >99% of all benign CNVs are inherited, and the vast majority of inherited CNVs are much smaller than 500 kb [1]. Most CMA platforms—including Affymetrix, Agilent, Illumina, and the Phalanx CytoOneArray—have a resolution of 100 kb, and most follow the 400 kb interpretation cutoff recommended by the ACMG. Depending on the patient population (Table 3), the variable detection rate ranges from 2.15 to 39.79%, and reported pathogenic and likely pathogenic CNVs vary from 0.38% to 8.27%. In accordance with the guidelines, the CMA analysis in this study achieved a 2.89% (30/1037) higher detection rate for reported pathogenic and likely pathogenic CNVs ≥ 400 kb and a 3.57% (37/1037) higher rate for nonreported CNVs. When including the 42 cases of aneuploidy and 43 cases with CNVs < 400 kb, the overall detection rate was 14.7% (152/1037).

### 4.2. Detection of Uniparental Disomy by CGH and SNP Platforms

Most occurrences of uniparental disomy (UPD) in chromosomes do not result in phenotypic anomalies. Maternal UPD involving chromosomes 2, 7, 14, and 15 and paternal UPD involving chromosomes 6, 11, 15, and 20 are associated with phenotypic growth, neurodevelopmental, and behavioral abnormalities [27,34]. UPD in maternal chromosome 7 is associated with a phenotype similar to Russell–Silver syndrome with intrauterine growth restriction [35]. As shown in the pathogenic and likely pathogenic CNVs in Table 2, case 9 exhibited a de novo 11.45 Mb gene deletion at the 7q21.11q21.3 loci related to Russell–Silver syndrome; both parents had a normal chromosome 7. In actuality, aCGH cannot determine the UPD condition in a patient; however, case 7 shows that a loss of copy has the potential to result in an isodisomy-related phenotype. Unfortunately, the methylation analysis is not available and we lack data to rule out the disorder. As this imprinting disorder is characterized by pre- and postnatal growth retardation; a triangular face; and facial, limb or truncal asymmetry, the parents decided to terminate the pregnancy. Another case exhibited a maternally inherited 3.805 Mb 7q21.12q21.2 duplication classified as a VOUS. Because the risk of heterodisomy or isodisomy is very low in a case with the gain of imprinting loci, the parents decided to continue the pregnancy, and a healthy baby was born (Appendix A). Neither of these cases met the criteria for UPD.

### 4.3. Does CMA Provide a Higher Detection YIELD Than Karyotyping?

In this study (Figure 1), all 42 cases with aneuploidy, including trisomy 21 (*n* = 14), 18 (*n* = 10), and 13 (*n* = 3); monosomy X (*n* = 5); 47,XXY (*n* = 3); 47,XXX (*n* = 2); and other variants (*n* = 5), were diagnosed with both prenatal G-banding karyotyping and CMA. However, six balanced translocations, four inversions, and six low-percentage mosaicisms were not detected with CMA. The aCGH used in this article can also detect the mosaic level above 60%. The overall detection rate of CMA was 14.65% (152/1037). Despite the aneuploidies, the incremental yield of 10.6% (110/1037) included 67 cases with CNVs ≥ 400 kb (10.6%; 67/1037), including cases with reported pathogenic or likely pathogenic variants (2.9% (30/1037)) and cases with nonreported VOUS, benign, or likely benign variants (3.6% (37/1037)). Moreover, cases with nonreported CNVs < 400 kb were also detected by CMA (4.1% (43/1037)). According to our review, CGH + SNP almost dominates the current prenatal testing platform. CGH + SNP can provide more clinical information on genetic findings, such as smaller CNV, UPD or tetraploidy, etc., and those findings provide valuable information for the consideration of possible clinical outcomes. However, for some findings, further specific tests are necessary to confirm the diagnosis; for instance, a methylation test is still necessary to rule out whether the loss of imprinting expressed in the fetus results in UPD disorder; thus, it still a difficult and urgent condition in prenatal diagnosis. We recognize that the CGH platform may not provide the information as the SNP array does. In order to make up for this shortcoming, our strategy is to conduct the karyotyping and aCGH side by side, and the results show that targeted aCGH can indeed help discover most of the significant CNVs to improve the diagnosis rate.

### 4.4. Does CMA Have a Higher Detection Yield for Fetuses with Increased NT and cFTS?

RCP/RCOG/BSGM recommendations indicate that CMA should be performed if nuchal translucency (NT) ≥ 3.5 mm occurs when the crown–rump length measures between 45–84 mm [21]. A systematic review and metanalysis of 17 studies reported that genomic microarrays provided a 5.0% incremental yield for detecting CNVs and aberrations, including those involving 22q11.2, in fetuses with isolated increased NT and normal karyotypes [36]. Lund et al. [37] reported a detection rate of 12.8% in a prospective clinical series using high-resolution prenatal CMA on uncultured CVS. The detection rate of clinically important chromosomal anomalies was significantly higher in pregnancies with NT ≥ 3.5 mm. In pregnancies with NT ≥ 4.5 mm, the detection rate was as high as 26.5% [38]. In this study, CMA detected CNVs ≥ 400 kb in 33.3% (10/30) of cases with NT ≥ 3.5 mm. In cFTS risk assessment, when combining ultrasound soft markers (abnormal doppler flow of the ductus venosus or tricuspid and an absent nasal bone) and serum in the evaluation, the major indication for high-risk screening of CMA was revealed to be 43.3% (13/30), which is a significantly higher detection rate (Table 2). This means that NT and cFTS in CMA are more effective than other single soft markers.

### 4.5. Relationship between CMA and Ultrasound Scan Abnormalities

Congenital anomalies are highly correlated with chromosomal abnormalities, and vary depending on the number and type of scan anomalies. De Wit reported that pathogenic CNVs can be detected in 5.6% of fetuses with isolated anomalies and 9.1% of fetuses with multiple anomalies [16]. In this study (Table 1), the major indications in abnormal ultrasounds were omphalocele, micrognathia, radial aplasia, megacystis, pyelectasis, ambiguous genitalia, echogenic bowel, VACTERL, and heart defects; one or more of these features was detected among 36.7% (11/30) of fetuses with reported pathogenic and likely pathogenic CNVs ≥ 400 kb. Thus, CMA obviously increases the detection yield in prenatal diagnosis.

CMA is recommended for all types of CHD in cases with prenatally diagnosed fetal cardiovascular malformations [38,39,40]. Combined CMA and next-generation sequencing detected pathogenic chromosomal anomalies in 21 of 115 (18.3%) fetuses with CHDs [10]. As shown in Table 2, CMA detected two cases with reported pathogenic and likely CNVs ≥ 400 kb associated with pathogenic congenital heart abnormalities. Case 14 had a double outlet of the ventricle (10q23 deletion syndrome; MIM: 612242) and case 16 had tetralogy of Fallot (16p11.2 deletion syndrome; MIM: 611913); both of these cardiovascular malformations are associated with aberrations in CMA, which suggests that CMA may enable more detailed analysis for the detection of CHD.

### 4.6. CNVs and Advanced Maternal Age

The aneuploidy abnormality rate increases with AMA. CMA has been reported to increase the prenatal diagnostic rate by about 1.4–1.7% for pregnancies in women with AMA [4,26]. In this study (Table 1), AMA was the major indication for CMA, and 0.55% (3/546) of the reported CNVs ≥ 400 kb were lower than those reported in the literature. However, in the nonreported CNVs ≥ 400 kb, 2.56% (14/546) were more common than those reported in the AMA group.

### 4.7. Prenatal Clinical Counseling for Inherited CNVs or CNVs with Variable Penetrance and Expressivity

Although the generalization between the size and significance of CNVs holds true as a general rule, very large CNVs can be benign in nature, and very small CNVs can be clinically significant [2]. To estimate the penetrance for recurrent pathogenic CNVs, the background risk for congenital anomalies/developmental delay/intellectual disability was assumed to be ~5%. RCOP/BSGM/RCOG recommendations for the use of CMA in pregnancy susceptibility includes distal 1q21.1 deletions and duplications, 15q13.3 deletions, distal and proximal 16p11.2 deletions, and 17q12 deletions. Detailed scans looking for associated anomalies in a clinical context should be considered when reporting these variations, including 22q11.2 duplication, proximal 1q21.1 deletion, and 17q12 duplication [21]. To compare the frequency of CNVs of variable penetrance in low-risk and high-risk prenatal samples, the CNVs were categorized based on clinical penetrance as: (i) high (>40%), (ii) moderate (10–40%), and (iii) low (<10%). High-penetrance CNVs play a major role in the overall heritability of developmental, intellectual, and structural anomalies. Low-penetrance CNVs do not seem to contribute to these anomalies [41].

Table 2 shows the cases with pathogenic and likely pathogenic CNVs ≥ 400 kb. Firstly, two cases of de novo proximal 16p 11.2 deletion CNVs involving the recurrent BP4 and BP5 breakpoint (BP) regions, including the morbid gene *TBX6* (MIM: 611913), were detected in this study; the deletion of this region may be associated with developmental delay, cognitive impairment, language delay, autism spectrum disorder, delayed language development, or minor dysmorphic facial features. In both of these cases, the parents decided to TOP due to the associated ultrasound anomalies and a high penetrance estimate of 46.8%. Case 18 presented with 16p 11.2 duplication with a moderate penetrance estimate of 27.2%; the parents decided to continue the pregnancy, and a live baby was born. Secondly, case 19 had maternally inherited 16p13.11 duplication affecting the morbid gene *MYH11* (MIM: 160745); cFTS risk evaluation revealed tricuspid regurgitation and the absence of a nasal bone, and intrauterine fetal death occurred at 32 weeks gestation. Thirdly, both case 20 (de novo) and case 21 (maternally inherited) involved 22q11.1q11.21 duplication affecting the morbid genes *TBX1* (MIM: 602054) and *CECR2* (MIM: 115470); furthermore, the duplication region partially overlapped with a critical region of cat eye syndrome (CES). CES is characterized by large phenotypic variability, ranging from near normal to severe malformations, as reflected by varied neurodevelopmental outcomes [42]. In general, CES is typically associated with a supernumerary bisatellited marker chromosome (inv dup 22pter-22q11.2), resulting in four copies of this region, but aCGH cannot well discriminate the difference between three and four copies; thus, we further checked the karyotype data of both cases. Case 20 showed a mosaic karyotype with a supernumerary marker chromosome (47,XY+ mar de novo [42]/46,XY [29]) and pyelectasis was found with an ultrasound. Therefore, the fetus was diagnosed as having CES, and, finally, the parents decided to TOP. However, the gain of CNV in case 21 was maternally inherited and the karyotype was normal. Some publications report that the interstitial duplication of 22q11.2 is associated with typical CES [43]. Based on those reports, the risk of pathogenicity cannot be ruled out, and it was considered to be a likely pathogenic case. The parents decided to continue the pregnancy, and a healthy baby was born; after three years following, no significant clinical issues were found. Fourthly, 22q11.21 duplication syndrome (MIM: 608363) was noted in two cases. Case 25 had increased NT (4.4 mm) and the variant was de novo; thus, the parents decided to TOP. Case 26 was maternally inherited; thus, the parents decided to continue the pregnancy, and a healthy baby was born [14]. Lastly, Xq28 duplication syndrome (MIM: 300815) affecting the morbid gene *RB39B*, associated with intellectual developmental disorder, was noted in case 25. The ultrasound scan revealed ambiguous genitalia, and the variant was de novo (MIM 300815); thus, the parents decided to TOP. In case 30, the fetal ultrasound scan revealed an echogenic bowel; in this case, the variant was paternally inherited, and a healthy baby was born.

When a CNV is found in a parent or another relevant family member, numerous caveats should be considered. Many genes may include variations related to secondary or incidental findings associated with adult-onset conditions or carrier status. It is recommended that solicited CNV pathogenic findings should only be reported when the identified variant may inform present or future management of the pregnancy or family; nonactionable findings should not be reported. Clinics and laboratories should report female carriers of X-linked recessive mutations associated with childhood-onset disorders, since there may be significant risk to the family if affected males are conceived [21]. In this study, one case with a maternally inherited CNV reported as Xp21.1 deletion was diagnosed as pathogenic Duchene muscular dystrophy (MIM: 310200), which affects the female carriers of X-linked recessive mutations associated with childhood-onset disorders. There may be significant risk to the family if affected males are conceived. As the fetus was female, we disclosed the heterozygous recessive findings. After thorough counseling, the parents decided to carry on with the pregnancy, and a healthy female was born.

The Appendix A shows all 37 cases with nonreported CNVs < 400 kb detected by CMA and interpreted as VOUS, benign, or likely benign for all types of inheritance. There is evidence to refute the significance of the duplication of the Xp22.31 (STS) region. Duplications of this region are common in the general population (~0.32–0.41%). Recent studies on carriers of this duplication, identified through large cohort studies of the general population, showed that carriers perform similarly in neurocognitive tests to noncarrier controls [44]. However, recessive X-linked ichthyosis (MIM: 308100) is carried in females and only phenotypically manifests in males. Firstly, eight cases of Xp22.31 duplication were detected in this study: one de novo deletion associated with pathogenic X-linked mental retardation and seven benign inherited variants; all parents decided to continue the pregnancies and healthy babies were born. During counseling to take family histories, most male family members reported ichthyosis, but the females were healthy. Secondly, the prevalence of Y-chromosome deletion or microdeletion is estimated to be one in 2000–3000 in males. The frequency of Yq microdeletions in males with azoospermia and oligozoospermia is about 5–15% [45]. This study detected eight cases with Yq11.223q11.23 variants, all paternally inherited (four deletions and four duplications) ranging in size from 0.415 to 2.246 Mb; we did not report all of these variants. Moreover, three and four cases of common CNVs ≥ 400 kb associated with 15q11.2 deletion and 15q13.3 duplication were detected, respectively. ClinGen queries of the Database of Genomic Variants (DGV) gold-standard GRCh37 dataset indicated no relationships have been reported between the gene(s) included in this region and human disease. Given the high population frequency, this region has been classified as “dosage sensitivity unlikely”.

Finally, providing information on incidental prenatal findings may have profound consequences, as the parents can opt to terminate the pregnancy. Many CNVs are associated with phenotypes that have reduced penetrance and/or variable expressivity. Some cases may inherit CNVs from the parents, but their penetrance varies dramatically, and ultrasound may show severe multiple fetal anomalies. Some women find this information to be “toxic” and to cause considerable amounts of anxiety during pregnancy and beyond, and even during their baby’s childhood [46]. The identification of a CNV allows for a more precise understanding of the medical and neurocognitive implications of the anomaly, which is important when making decisions about the pregnancy and in planning care for the child [18]. In this study, the chromosomal microarray technologies standard was followed up with the ACMG update guideline in order to provide good quality in the clinical application of the diagnostic evaluation of constitutional disorders [47]. Studies using pre- and post-tests have described the potential limitations of CMA as a clinical test, especially with regard to the detection of VOUS, balanced translocations, and low-level mosaicism. The reporting of VOUS to families in a prenatal setting should be discussed in the context of VOUS detected by CMA.

In conclusion, this study demonstrates the clear value of CMA for the evaluation of cFTS and fetal structural anomalies. The variable clinical implications of inherited CNVs also depend on the mechanisms of inheritance that influence the expression of a trait. Genetic counseling and the evaluation of the risk of recurrence of the genetic abnormality are important for families. However, larger prospective cohort studies with greater focus on additional information are needed.

## 5. Conclusions

Prenatal CMA is recommended for high-risk pregnancies.

## Figures and Tables

**Figure 1 jcm-11-03624-f001:**
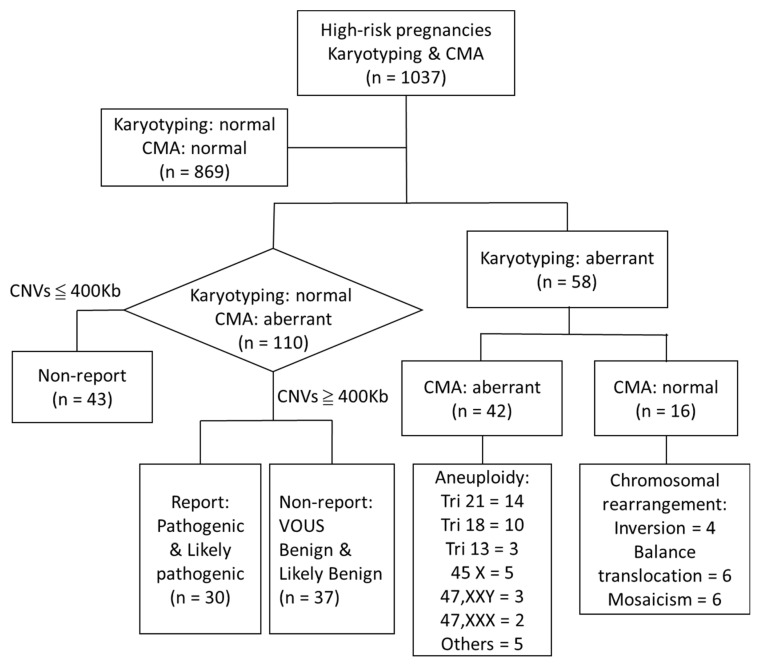
Flowchart of patient G-banding karyotype and chromosomal microarray analysis. CMA: chromosomal microarray (CMA); CNVs: copy number variants; VOUS: variant of uncertain significance.

**Table 1 jcm-11-03624-t001:** Major indications for prenatal chromosomal microarray analysis (CMA) for 1037 pregnancies with aberrant findings.

Indication	No	T21	T18	T13	Reported CNVs ≥ 400 kb	Nonreported CNVs ≥ 400 kb
Advanced maternal age only	546 (52.7%)	1	1	3	3	14
Abnormal ultrasound finding	197 (19.0%)	2	4	1	11	2
High risk on Down syndrome screening (≥1:270)	189 (18.2%)	13	10	3	13	9
Increased NT (≥3.5 mm) only	90	6	5	2	1	3
FTS soft markers: DV(+), TR(+), NB(−)	56	7	5	1	1	4
Previous pregnancy, child, or familial risk	105 (10.1%)				1	5

T21: trisomy 21; T18: trisomy 18; T13: trisomy 13; CNVs: copy number variants; NT: nuchal translucency; FTS: first-trimester screening; DV(+): ductus venosus feverse; TR(+): tricuspid regurgitation; NB(−): absent nasal bone.

**Table 2 jcm-11-03624-t002:** Overview of the pathologic, likely pathologic, and susceptibility copy number variants reported after chromosome microarray analysis of pregnancies in this study.

Case	Age	NT (mm)	Indication/Ultrasound Finding	CMA [hg19] Results	Size (Mb)	Inheritance	Candidate OMIM Genes	Disorders	Interpretation	Outcome
1	34	2.5	TR(+), High risk	1q21.1q21.2 (146,627,038–147,384,032) × 1	0.757	De novo	612474	1q21.1 deletion syndrome	Pathogenic	TOP
2	35	3.5	NT(+), High risk	2q37.2q37.3 (236,330,093–243,040,324) × 1	6.710	De novo	600430	2q37 deletion syndrome	Pathogenic	TOP
3	36	3.2	Omphalocele, High risk	3p26.33p26.2 (2,146,782–3,771,742) × 3	1.625	Paternal	607280	3pter-p25 duplication	VOUS	TOP
4	21	2.2	CM(+), TR(+), NB(−),High risk	3q22.1q25.32 (130,521,560–157,015,801) × 1	26.494	De novo	220200	Syndromic intellectual disability Dandy Walker Syndrome	Pathogenic	TOP
5	38	5.1	NT(+), DV(+), High risk, AMA	3q27.2q29 (184,799,629–197,803,820) × 3 9p24.3p22.3 (271,257–14,680,180) × 1 9p22.3p13.1 (14,844,795–38,663,271) × 3	13.004 14.409 23.818	De novo	611936/602424/604935/612900/158170/608980/156540/601673	3q29 duplication syndrome 9p24.3p22 deletion 9p22.3p13.1 duplication 9p duplication & deletion	LP Pathogenic	TOP
6	27	2.8	High risk	4q34.3q35.1 (180,742,112–183,532,267) × 3 4q35.1q35.2 (183,532,267–190,957,460) × 1	2.790 7.425	De novo	610083/518900	4q34.3q35.1 duplication 4q35.1q35.2 deletion	LP	TOP
7	25	4	Micrognathia, Low set ear, NT(+)	5q32 (145,755,389–150,297,954) × 1	2.575	De novo	N/A	Treacher Collins syndrome	Pathogenic	TOP
8	36	2.8	TR(+), High Risk, AMA	6q22.1q22.31 (115,853,923–119,245,348) × 3	3.391	De novo	605942/604714/612647/612659/610463/618865/172405/610098/120110	6q22.1q22.31 duplication	VOUS	LB (MR)
9	37	2	AMA	7q21.11q21.3 (84,600,949–96,051,291) × 1	11.45	De novo	600028/604149	Split-hand/foot malformation 1 Silver–Russell syndrome Myoclonus-Dystonia	Pathogenic	TOP
10	36	2.6	High risk, AMA	8q23.1 (106,336,068–106,715,982) × 1	0.380		603693	congenital diaphragmatic hernia	LP	BH
11	35	5.1	NT(+), AMA	9p24.3p22.2 (204,193–16,626,507) × 1	16.422	De novo	158170	9p24.3p22.2 deletion syndrome	Pathogenic	TOP
12	31	5.7	Radial aplasia, NT(+), High risk	9q21.2 (80,191,465–80,601,045) × 3	0.41	De novo	600998	Radial aplasia	VOUS	TOP
13	36	1.9	High risk, AMA	10q22.3 (79,617,635–81,707,527) × 3	2.090	De novo	602412/614258/607159/178642/178630/618639	10q22.3 duplication	VOUS	BH
14	32	1.4	Double outlet of right ventricle, TR(+)	10q23.1 (86,767,729–86,984,308) × 1	0.217	De novo	N/A	10q23 deletion syndrome	VOUS	TOP
15	35	4.5	NT(+), High risk, AMA	14q32.31q32.33 (101,758,166–106,852,173) × 1	5.094	De novo	614062/605799/614730	14q32.31q32.33 deletion syndrome	Pathogenic	TOP
16	27	1.9	TOF of Heart, NB(−), High risk	16p11.2 (29,653,115–30,198,522) × 1	0.545	De novo	611913	Proximal 16p11.2 deletion syndrome	Pathogenic	TOP
17	31	4.3	R/O: VACTERL, NT(+), High risk	16p11.2 (29,653,115–30,198,581) × 1	0.545	De novo	611913	Proximal 16p11.2 deletion syndrome	Pathogenic	TOP
18	32	1.5	Megacystis	16p11.2 (29,698,283–30,198,582) × 3	0.500	De novo	614671	Proximal 16p11.2 duplication syndrome	Pathogenic	BH
19	37	2.3	TR(+), NB(−), High risk	16p13.11 (15,131,575–16,288,874) × 3	1.157	Maternal	609449/160745/603234	16p13.1 duplication	LP	IUFD
20	40	1.7	Pyelectasis	22q11.1q11.21 (17,444,646–18,106,018) × 3	0.661	De novo	115470	Cat eye syndrome (47,XY+ mar de novo [42]/46,XY [29])	Pathogenic	TOP
21	37	2	AMA	22q11.1q11.21 (17,444,646–17,993,089) × 3	0.548	Maternal	115470	22q11.1q11.21 duplication (46,XY)	LP	BH
22	31	9.4	NT(+), NB(−), TR(+), High risk	22q11.21 (19,035,231–21,449,413) × 1	2.414	De novo	188400	DiGeorge Syndrome	Pathogenic	TOF
23	42	4.3	NT(+), TR(+), High risk, AMA	22q11.21 (19,006,943–21,461,068) × 1	2.454	De novo	188400	DiGeorge syndrome	Pathogenic	TOP
24	42	2.5	TR(+), High risk, AMA	22q11.21 (19,006,943–21,461,005) × 1 17q12 (34,823,708–36,247,940) × 3	2.454 1.426	De novo	614526 188400	DiGeorge syndrome 17q12 duplication syndrome	Pathogenic	TOP
25	33	4.4	NT(+)	22q11.21 (18,104,691–21,461,005) × 3	3.356	De novo	608363	22q11.2 duplication syndrome	Pathogenic	TOP
26	35	2.1	PHx; Genetic Hx	22q11.21 (19,006,943–21,461,005) × 3	2.454	Maternal	608363	22q11.2 duplication syndrome	Pathogenic	BH (Health)
27	40	2.3	AMA	Xp22.31 (6,460,120–8,101,239) × 1	1.641	De novo	308100	X-linked mental retardation Ichthyosis, X-linked (XLI)	Pathogenic	TOP
28	31	1.4	NB(−), Mental retardation	Xq22.1q22.2 (100,907,854–102,659,284) × 1	1.751	De novo	300319/300969	Xq22.1q22.2 deletion X-linked mental retardation	LP	LB (MR)
29	29	2.1	Ambigious genital	Xq28 (154,130,347–154,527,746) × 3	0.397	De novo	300815	Xq28 duplication syndrome	Pathogenic	TOP
30	31	2.5	Echogenic bowel	Xq28 (154,161,678–154,650,677) × 3 7q36.2 (153,923,581–154,024,097) × 1	0.351 0.087	Paternal Maternal	300815 612956	Xq28 duplication syndrome 7q36.2 deletion syndrome	Pathogenic VOUS	BH

NT: nuchal translucency; NB(−): absent nasal bone; TR(+): tricuspid regurgitation; DV(+): ductus venosus reverse; CMA: chromosome microarray; OMIN: Online Mendelian Inheritance in Man; VOUS: variants of uncertain significance; TOP: termination of pregnancy; LP: likely pathogenic; LB: live born; BH: born healthy.

**Table 3 jcm-11-03624-t003:** Overview of current CMA platforms and interpretation of prenatal examinations in the literature.

Author	Patient Population	Cases No	CMA Platform	Chip Design	CMA Resolution	Interpretation Cut Off	Detection Rate	P/LP CNVs
Oneda et al. (2014) [23]	High risk ^#^	464	Affymetrix cytogenetics Whole Genome 2.7 M array/Cytoscan HD Array	CGH + SNP	20–100 Kb	20–100 Kb	17/464 (3.70%)	15/464 (3.23%)
Zhu et al. (2016) [5]	Heart anomaly	115	Affymetrix CytoScan 750 K	CGH + SNP	100 Kb	N/A	21/115 (18.3%)	13/115 (11.3%)
Egloff et al. (2018) [24]	High risk ^#^	599	Agilent PreCytoNEM	CGH + SNP	60 &180 Kb	N/A	53/599 (8.85%)	16/599 (2.67%)
Sagi-Dain et al. (2018) [25]	Ultrasound anomaly	5750	Affymetrix CytoScan 750 K array Infinium OmniExpress-24 v1.2 BeadChip BlueGnome Cytochip ISCA 8 × 60 K format Agilent CGH + SNP (4 × 180 K)	CGH + SNP SNP CGH CGH + SNP	100 Kb	1 M (loss)/2 M (gain)	272/5750 (4.73%)	157/5750 (2.73%)
Vogel et al. (2018) [7]	cFTS high risk	575	Agilent CytoGenomics	CGH + SNP	180 K	N/A	51/575 (8.87%)	15/575 (2.61%)
Shi et al. (2019) [26]	High risk, AMA	703	Affymetrix CytoScan 750 K	CGH + SNP	100 Kb	N/A	48/703 (6.83%)	10/703 (1.42%)
Wang et al. (2019) [27]	High risk ^#^	5026	Affymetrix Human SNP Array 6.0 Affymetrix CytoScan HD	SNP CGH + SNP	Target: 20 kb (loss)/100 kb (gain) Nontarget: 50 kb (loss)/200 kb (gain)	400 K	562/5026 (11.2%)	19/5026 (0.38%)
Lin et al. (2020) [28]	General population	10,377	Thermo-Fisher CytoScan750 K	CGH + SNP	100 Kb	200 K	223/10,377 (2.15%)	126/10,377 (1.21%)
Xia et al. (2020) [29]	Ultrasound anomaly	477	Affymetrix CytoScan 750 K	CGH + SNP	50 Kb (loss)/100 Kb (gain)	100 K (loss)/200 K (gain)	71/447 (15.88%)	17/447 (3.80%)
Hu et al. (2021) [30]	Ultrasound anomaly	2466	Thermo-Fisher CytoScan750 K	CGH + SNP	100 Kb	400 K	107/2466 (4.34%)	64/2466 (2.59%)
Hu et al. (2021) [31]	AMA, soft marker	1521	Affymetrix CytoScan 750 K	CGH + SNP	100 Kb	400 K	330/1527 (21.61%)	37/1520 (2.42%)
Stern et al. (2021) [32]	Ultrasound low risk	6431	Affymetrix CytoScan 750 K	CGH + SNP	100 Kb	N/A	319/6431 (4.96%)	27/6431 (0.42%)
Wu et al. (2021) [33]	Serum screening high risk	713	Affymetrix CytoScan 750 K	CGH + SNP	100 Kb	400 K	82/713 (11.5%)	59/713 (8.27%)
Zhu et al. (2021) [10]	High risk ^#^	774	Affymetrix CytoScan 750 K Iliumina HumanCytoSNP-12 Agilent CGH 8 × 60 K (customized)	CGH + SNP SNP CGH + SNP	100 Kb	400 K	308/774 (39.79%)	17/774 (2.20%)
Present study	High risk ^#^	1037	Phalanx CytoOne	CGH	Target: 50–100 K; Non-target: 1 Mb	400 K	153/1037 (14.75%)	30/1037 (2.89%)

High risk ^#^: includes trisomy, CNVs; high risk with AMA, structural abnormalities on ultrasound, screening high risk, family history.

## Data Availability

The data are not publicly available as they contain information that could compromise the privacy of the research participants, but encoded data are available from the corresponding author upon reasonable request.

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
