# Peer review of "Prenatal Diagnosis Using Chromosomal Microarray Analysis in High-Risk Pregnancies"

_jcm, 2022, doi:10.3390/jcm11133624_

Round 1

Reviewer 1 Report

The Authors report a prospective cohort study of 1037 high-risk pregnancies that underwent invasive prenatal diagnosis using conventional karyotyping and  CMA.

·         General comments
The manuscript highlight the relevance of CMA technology in prenatal diagnosis confirming results from previous recent articles which should be cited .

In general, the manuscript is clear, of some relevance for the field and presented in a well-structured manner; the figures and the tables appear appropriate, properly show the data which are easy to interpret and understand.

In Materials and Methods more details are needed about DNA sample handling and aCGH platform used.

In Results section, data from prenatal diagnosis for AMA could be clearly separated from the significant remaining results.

Despite interesting, Discussion could be summarized, especially in the Prenatal counselling section. However, in pregnancies with scan anomalies discussion about CMA clinical relevance could be improved, by comparing further CMA results with literature data.

·         Specific comments

In Table 1 some acronyms should be explained in footnotes.

Some errors in karyotype description should be amended.

Author Response

The Authors report a prospective cohort study of 1037 high-risk pregnancies that underwent invasive prenatal diagnosis using conventional karyotyping and CMA.

  • General comments
    Point 1: The manuscript highlights the relevance of CMA technology in prenatal diagnosis confirming results from previous recent articles which should be cited.

In general, the manuscript is clear, of some relevance for the field and presented in a well-structured manner; the figures and the tables appear appropriate, properly show the data which are easy to interpret and understand.

 Response 1:

Thanks for the comments. Manuscript will be revised according to comments from reviewers.

Point 2: In Materials and Methods more details are needed about DNA sample handling and aCGH platform used.

Response 2:

We have revised materials and methods with more details.

Chorionic villus sampling (CVS) and amniocentesis were performed for G-banding karyotype analysis and CMA. Before DNA extraction, CVS samples were dissected under a microscope to remove any contaminating maternal decidua. DNAs were immediately extracted from the uncultured villi (10–20 mg) and amniocentesis samples (5-10 ml) using Norgen's Blood Genomic DNA Isolation Mini Kit (Norgen Biotek Corp., Thorold, Ontario, Canada).

For CMA, all samples were tested on targeted arrays using the CytoOneArray® array platform (versions 1.0 to 4.0), depending on the date of specimen receipt as the design of these arrays changed over time (Phalanx Biotech Group Inc., Taiwan, ROC) [11]. The targeted design of CytoOneArray® can analyze more than 500 DNA regions associated with known genetic disorders, especially in Asia populations (https://www.phalanx.com.tw/reproductive-medicine/prenatal-testing/cyto-390/?lang=en).

Point 3: In Results section, data from prenatal diagnosis for AMA could be clearly separated from the significant remaining results.

Response 3:

 Thanks for the suggestions. We have added more explanation regarding AMA in results section.

In page 5: As shown in Table 1, among the 546 cases (52.7%; 546/1037) cases with AMA only is the major indication for prenatal testing with CMA (52.7%; 546/1037). 4% (22/546) of them showed chromosomal abnormality with CMA. However, only 1.1% (6/546) showed pathogenic/likely pathogenic CNVs. It indicated that AMA only is less relevant than other indications in Table 1.

Point 4: Despite interesting, Discussion could be summarized, especially in the Prenatal counselling section. However, in pregnancies with scan anomalies discussion about CMA clinical relevance could be improved, by comparing further CMA results with literature data.

Response 4:

 Thanks for the suggestion. We have revised discussion 4.7 with more explanation regarding CMA clinical relevance.

Pregnancies with scan anomalies usually caused considerable amounts of anxiety for parents. CMA clearly improved the diagnosis yields (9.1%; 18/197) in this group and provide more information for parents’ decision making. In this study, we have similar finding regarding CMA diagnostic improvement by comparing other CMA results with literature data.

  • Specific comments

Point 5: In Table 1 some acronyms should be explained in footnotes.

Some errors in karyotype description should be amended.

Response 5:

Thanks for suggestions. The table 1 added DV, TR, NB acronyms. It is revised the Table. Besides, the errors karyotype also corrected.

Reviewer 2 Report

The article of C.H. Hsiao and colleagues, Prenatal diagnosis using chromosomal microarray analysis in high-risk pregnancies reports a prospective research study of 1037 high risk pregnancies presenting for invasive diagnostic genetic testing at a single institution. The study evaluated use of chromosomal microarray vs G-banded karyotyping for detection of chromosomal aneuploidies and submicroscopic copy number aberrations, ie deletions and duplications. The article is well written and follows guidelines for testing that are endorsed by SMFM and ACOG and are largely already in clinical use based on a larger prior study by Wapner et al (2012; ref 4). Unlike that initial study over a decade ago, where multiple array platforms were used and often difficult to compare, this study used a uniform array platform and did make comparisons across currently used array modalities in Table 3, which is helpful.

There are areas of the article that would benefit from clarification:

1.    Currently, the most prevalent constitutional array platform used for both postnatal and prenatal applications appears to be the Cytoscan HD CGH + SNP approach. There are differences in detection when comparing between a CGH only array as was used here vs those with SNPs. This is not addressed in the article. In particular:

a.    There can be differences in the log2 ratio between aCGH and SNP arrays and this is especially important for mosaic cases. The authors did not indicate the degree of mosaicism that was detectable by aCGH compared to karyotype but many SNP-based arrays can detect mosaicism in the 20-30% range and below. This should be clarified.

b.    Utilizing the SNP probes can also help with ploidy issues, the authors state up front that triploidy cannot be readily detected by CMA, however as is demonstrated in Figure 4 of ref 3, this is no longer the case when using SNP-based arrays. The statement at the beginning discussing the limitations of CMA should be qualified.

c.     The aCGH platform used for this study has approximately 33,000 probes across 331 regions that are enriched for known copy number variation. The more commonly used Cytoscan HD has 2.6 million probes including SNPs and provides a more “objective” analysis, there are drawbacks to that approach such as increased VUS that may be worth discussing further since this point may make the current study distinct from previously published work, ie, why was this array platform used?

2.    In the reference to UPD in section 4.2, the concept of isodisomy which would be detected particularly on SNP-based arrays vs that of heterodisomy, which will not likely be detected, should be clarified. This is why methylation analysis for PWS/AS is still indicated today.

3.    The concept of runs of homozygosity and identity by descent that may indicate consanguinity is not an infrequent finding in CMA analysis and should be addressed in the article

4.     In the section regarding US findings and congenital heart defects, it may be worth noting that one of the major causes of an increased NT is a RASopathy such as Noonan syndrome and the molecular abnormalities that cause Noonan syndrome are not detectable by CMA since they are primarily single nucleotide changes found on sequencing.

5.    Is the term VACTAL used interchangeably for VACTERL association? If not, please define it.

6.    In Table 2 and in the text, the disorders discussed are rushed and are confusing.  For example, the 22q11.2 duplication syndrome is 3 copies of the region that involves the same one as DiGeorge/Velocardiofacial syndrome (when deleted) and typically encompasses the region from 18,000,000-21,000,000 including the TBX1 and CRKL genes. Instead, the Cat Eye region is proximal to this and is smaller as appears to be seen in Case 21 where the coordinates point to a diagnosis of Cat Eye syndrome, however, CES is known to be 4 copies of that region and is often due to the presence of a bisatellited marker chromosome 22q. This should be clarified and perhaps the data assessed as to copy number. Another read-through of that entire section that discusses the Table, (4.7) is advised. 

7.  There is a statement regarding regions >400 kb in DGV with no associated clinical findings. Please clarify that those in 15q11.2 are not the regions associated with Prader-Willi and Angelman syndrome or the 15q11.2 duplication syndrome.

8.    Please update the ACMG technical standards reference and guidelines to that from 2021: Shao et al, Genetics in Medicine, 2021

Finally, while it is highly feasible to perform parental CMA analyses to help resolve VUS in the research setting, it is very challenging to do so in actual clinical practice, particularly given the timing constraints for TOP in many countries.

Author Response

There are areas of the article that would benefit from clarification:

Point 1.  Currently, the most prevalent constitutional array platform used for both postnatal and prenatal applications appear to be the Cytoscan HD CGH + SNP approach. There are differences in detection when comparing between a CGH only array as was used here vs those with SNPs. This is not addressed in the article. In particular:

  1. There can be differences in the log2 ratio between aCGH and SNP arrays and this is especially important for mosaic cases. The authors did not indicate the degree of mosaicism that was detectable by aCGH compared to karyotype but many SNP-based arrays can detect mosaicism in the 20-30% range and below. This should be clarified.

Response a. Thanks for the comments.

Here, our goal is to establish a quick, effective, reliable, and suitable screening flow that can be implemented in our daily prenatal screening for high-risk pregnancies. We learn each prenatal test has its benefit and limitation. In Taiwan, ultrasound and karyotyping are routine tests in high-risk pregnancies, in general, ultrasound can provide image information, and karyotyping can examine particular genetic aberrant including triploid, balance-translocation, aneuploidy, mosaicism, and so on. Based on our clinical experience and evaluation, the application of targeted aCGH already can further provide us with the lack of information on microdeletion/microduplication, in addition, a well-design targeted aCGH can fit the clinical needs to screening the genetic disorders with high prevalence presented in the local populations, and can increase the efficiency of analysis and counseling. In our design, the mosaicism is mainly diagnosed by karyotyping, however, the aCGH used in this article also can detect the mosaic level above 60%.

  1. Utilizing the SNP probes can also help with ploidy issues, the authors state up front that triploidy cannot be readily detected by CMA, however as is demonstrated in Figure 4 of ref 3, this is no longer the case when using SNP-based arrays. The statement at the beginning discussing the limitations of CMA should be qualified.

Response b. Thanks for the reviewer's comments.

Triploidy indeed can't detect by aCGH, so we apply the karyotyping to make up for the shortage.

  1. The aCGH platform used for this study has approximately 33,000 probes across 331 regions that are enriched for known copy number variation. The more commonly used Cytoscan HD has 2.6 million probes including SNPs and provides a more “objective” analysis, there are drawbacks to that approach such as increased VUS that may be worth discussing further since this point may make the current study distinct from previously published work, ie, why was this array platform used?

Response c: Thanks for the reviewer's comments.

For this question, we have explained in response a of point 1.

Point 2.  In the reference to UPD in section 4.2, the concept of isodisomy which would be detected particularly on SNP-based arrays vs that of heterodisomy, which will not likely be detected, should be clarified. This is why methylation analysis for PWS/AS is still indicated today.

 Response 2. Thanks for the reviewer's suggestions.

We have revised the paragraph below.

Most occurrences of uniparental disomy (UPD) in chromosomes do not result in phenotypic anomalies. Maternal UPD involving chromosomes 2, 7, 14, and 15 and paternal UPD involving chromosomes 6, 11, 15, and 20 are associated with phenotypic growth and behavior abnormalities [27, 35]. These phenotypic effects may be related to imprinting and most segmental UPD greater than 10 Mb from exome data. UPD of maternal chromosome 7 is associated with a phenotype similar to Russell-Silver syndrome with intrauterine growth restriction [36]. As shown in the pathogenic and likely pathogenic CNVs in Table 2, case 9 exhibited a de novo 11.45 Mb gene deletion at the 7q21.11q21.3 loci related to Russell-Silver syndrome; both parents had a normal chromosome 7. Actually, aCGH cannot determine the UPD condition in a patient, however, case 7 shows a loss of copy has the potential to result in an isodisomy-related phenotype. Unfortunately, the methylation analysis is not available and lack of data to rule out the disorder. However, as this imprinting disorder is characterized by pre- and postnatal growth retardation, a triangular face, and facial, limb or truncal asymmetry, the parents decided to terminate the pregnancy. Another case exhibited a maternally inherited 3.805 Mb 7q21.12q21.2 duplication classified as a VOUS. Because the risk of heterodisomy or isodisomy is very low in a case with the gain of imprinting loci. The parents decided to continue the pregnancy and a healthy baby was born (supplementary table). Neither of these cases met the criteria for UPD.

Point 3.  The concept of runs of homozygosity and identity by descent that may indicate consanguinity is not an infrequent finding in CMA analysis and should be addressed in the article

Response 3: Thanks for the reviewer's comments.

Actually, aCGH cannot provide genotyping data to call the region of the identity by descent to understand consanguinity.

Point 4. In the section regarding US findings and congenital heart defects, it may be worth noting that one of the major causes of an increased NT is a RASopathy such as Noonan syndrome and the molecular abnormalities that cause Noonan syndrome are not detectable by CMA since they are primarily single nucleotide changes found on sequencing.

 Response 4: Thanks for the reviewers remind.

In the case of specific US findings or very thick NT associated the Noonan syndrome that unable detected by CMA. We will use the sequencing to prove it.

Point 5. Is the term VACTAL used interchangeably for VACTERL association? If not, please define it.

Response 5:  VACTAL VACTERL

Point 6. In Table 2 and in the text, the disorders discussed are rushed and are confusing. For example, the 22q11.2 duplication syndrome is 3 copies of the region that involves the same one as DiGeorge/Velocardiofacial syndrome (when deleted) and typically encompasses the region from 18,000,000-21,000,000 including the TBX1 and CRKL genes. Instead, the Cat Eye region is proximal to this and is smaller as appears to be seen in Case 21 where the coordinates point to a diagnosis of Cat Eye syndrome, however, CES is known to be 4 copies of that region and is often due to the presence of a bisatellited marker chromosome 22q. This should be clarified and perhaps the data assessed as to copy number. Another read-through of that entire section that discusses the Table, (4.7) is advised. 

Response 6: Thanks for the reviewer's suggestions.

Table 2 summarized the results of the aCGH analysis in this article and listed the potential disorder correlated with identified regions. Those data are subject to further evaluation combined with all clinical observation and examination. For interpretation of aCGH results, the detected DNA regions were further checked according to the records of Decipher database, for instance, the deletion regions examined in case 22, 23, and 24, cover the defined regions of Velocardiofacial / DiGeorge syndrome. The gain of DNA region in case 20 and 21, partially encompass the defined region of Cat-Eye Syndrome (Type I). Because of technical limitations, aCGH can examine the gain of copies, but it is hard to distinguish whether to have 3 copies or 4 copies for example in CES. Taken together, those results highlight the potential disorder risks.

Point 7. There is a statement regarding regions >400 kb in DGV with no associated clinical findings. Please clarify that those in 15q11.2 are not the regions associated with Prader-Willi and Angelman syndrome or the 15q11.2 duplication syndrome.

Response 7: Thanks for the reviewers remind.

Three distinct neurodevelopmental disorders arise primarily from deletions or duplications that occur at the 15q11-q13 locus: Prader-Willi syndrome (PWS), Angelman syndrome (AS), and 15q11-q13 duplication syndrome (Dup15q syndrome). Each of these disorders results from the loss of function or over-expression of at least one imprinted gene. In supplementary data, we found a gain of 15q13.3 region (15q13.3 duplication) in case 12 and 16, and a loss of 15q11.2(15q11.2 deletion) in case 15, the aberrant regions are not including the major causative genes of Prader-Willi and Angelman syndrome. Thanks for the reviewers remind this point. We may aim to this locus for further study in the future.

Point 8.  Please update the ACMG technical standards reference and guidelines to that from 2021: Shao et al, Genetics in Medicine, 2021

Response 8:  Add this into we update and add it into the reference and citation.

Add section into page 11: In this study, the chromosomal microarray technologies standard was followed up ACMG the update guideline to provide good quality in clinical application in the diagnostic evaluation for constitutional disorders [45].

Point 9. Finally, while it is highly feasible to perform parental CMA analyses to help resolve VUS in the research setting, it is very challenging to do so in actual clinical practice, particularly given the timing constraints for TOP in many countries.

Response 9: Thanks for the reviewer's comments.

For VUS interpretation, it is a big challenge restricted by timing constraints and lack of enough information. However, it is no doubt parental CMA analysis is powerful to help rule out this issue, and it is almost a routine following action in Taiwan. The bigger data to clarify benign / pathogenic into minimize the percentage of VUS for clinical practitioner in future.

Round 2

Reviewer 2 Report

Review 2:

The authors have taken steps to address the issues raised in the initial review and the paper is improved. This is appreciated and their responses regarding the limitations of the platform are appropriate. There are still some minor issues that should be addressed as noted below:

Point 1a:  Thank you for addressing this. I do not believe your manuscript differentiated between screening and testing (diagnostic testing). Array CGH from an amnio and your study would be diagnostic testing rather than a population screen such as non-invasive prenatal screening using cell-free fetal DNA in the maternal circulation. I think it is clear in your manuscript but your response discusses prenatal screening, which I do not think is your intent. If it is, please clarify.

Point 2: suggestions for revised 4.2 in bold:

Most occurrences of uniparental disomy (UPD) in chromosomes do not result in phenotypic anomalies. Maternal UPD involving chromosomes 2, 7, 14, and 15 and paternal UPD involving chromosomes 6, 11, 15, and 20 are associated with phenotypic growth,neurodevelopmental and behavioral abnormalities [27, 35]. These phenotypic effects may be related to imprinting and most segmental UPD greater than 10 Mb from exome data. (????)[this may be incomplete]

UPD of maternal chromosome 7 is associated with a phenotype similar to Russell-Silver syndrome with intrauterine growth restriction [36]. As shown in the pathogenic and likely pathogenic CNVs in Table 2, case 9 exhibited a de novo 11.45 Mb gene deletion at the 7q21.11q21.3 loci related to Russell-Silver syndrome; both parents had a normal chromosome 7. Actually, aCGH cannot determine the UPD condition in a patient, however, case 7 shows a loss of copy has the potential to result in an isodisomy-related phenotype. Unfortunately, the methylation analysis is not available and we lack data to rule out the disorder. However, as this imprinting disorder is characterized by pre- and postnatal growth retardation, a triangular face, and facial, limb or truncal asymmetry, the parents decided to terminate the pregnancy. Another case exhibited a maternally inherited 3.805 Mb 7q21.12q21.2 duplication classified as a VOUS. Because the risk of heterodisomy or isodisomy is very low in a case with the gain of imprinting loci, the parents decided to continue the pregnancy and a healthy baby was born (supplementary table). Neither of these cases met the criteria for UPD.

Point 6. I will assume you have karyotyping that distinguishes the supernumerary bisatellited marker chromosome of Cat Eye (CES) with 4 copies of that region from a duplication of 22q11.2 which is 3 copies of a slightly more distal region. The phenotypes are quite different since CES can have significant cardiac and anal malformations. CES has been known to have 4 copies of the region since 1986 (PMID 3961499) also Mears 1994 (PMID 7912885). It stands out when looking at the Table that CES is listed as having 3 copies rather than the accepted 4. If the array cannot diagnose or differentiate between these two entities with distinct phenotypes, I would recommend either removing those cases or explaining why you have cases of CES with 3 copies of the region. If you have the karyotypes, that should address it. The more recent array platforms seem to do a better job with regions like this but since the platform used here targets this region, it should be accurately interpreted and diagnosed.

7. This response is perfect, thank you.

Three distinct neurodevelopmental disorders arise primarily from deletions or duplications that occur at the 15q11-q13 locus: Prader-Willi syndrome (PWS), Angelman syndrome (AS), and 15q11-q13 duplication syndrome (Dup15q syndrome). Each of these disorders results from the loss of function or over-expression of at least one imprinted gene. In supplementary data, we found a gain of 15q13.3 region (15q13.3 duplication) in case 12 and 16, and a loss of 15q11.2(15q11.2 deletion) in case 15, the aberrant regions are not including the major causative genes of Prader-Willi and Angelman syndrome.

Finally, on re-review, I would like to point out:

A)    For section 4.3 where the detection ability of this study and this single platform is discussed, it may be helpful for the reader if the authors also make a statement about the relative improved detection ability across the platforms that are analyzed in the Table.

B)    At the end of section 4.5 there is a speculative point about CMA and congenital heart defects. This is well recognized, I would recommend providing some references. For example: Landstrom et al, 2021, AHA statement; Wu et al 2017 PMD 5418813, and reviewed in Zaidi and Brueckner 2017, Circ Res

C)    In section 4.6, there is a statement about increased findings in AMA patients. Please clarify: while aneuploidy (e.g. trisomies) is well known to be associated with AMA.  However, in general are CNVs (submicroscopic, particularly those involving recurrent low copy repeat mediated CNVs such as those on chr 7q, 15q, 17, 22q etc) associated with AMA?? If this is the case, please provide references. If not, please modify.

D) Please review the last inserted revised section for grammar and writing flow

Author Response

The authors have taken steps to address the issues raised in the initial review and the paper is improved. This is appreciated and their responses regarding the limitations of the platform are appropriate. There are still some minor issues that should be addressed as noted below:

 Point 1a:  Thank you for addressing this. I do not believe your manuscript differentiated between screening and testing (diagnostic testing). Array CGH from an amnio and your study would be diagnostic testing rather than a population screen such as non-invasive prenatal screening using cell-free fetal DNA in the maternal circulation. I think it is clear in your manuscript but your response discusses prenatal screening, which I do not think is your intent. If it is, please clarify.

Response: Thanks for the reviewer's correction.

I apologize for making a misleading description of our goal in our response. CMA is a molecular genetic detection technology in the field of prenatal diagnosis, and it is no doubt we implement CMA in our routine prenatal diagnosis processes to enhance the diagnosis rate and efficiency in the clinic. I just want to emphasize that many testing tools are used in our clinical testing that can cover each other and provide more comprehensive information, and help deliver accurate prenatal diagnoses.

Point 2: suggestions for revised 4.2 in bold:

 Most occurrences of uniparental disomy (UPD) in chromosomes do not result in phenotypic anomalies. Maternal UPD involving chromosomes 2, 7, 14, and 15 and paternal UPD involving chromosomes 6, 11, 15, and 20 are associated with phenotypic growth,neurodevelopmental and behavioral abnormalities [27, 35]. These phenotypic effects may be related to imprinting and most segmental UPD greater than 10 Mb from exome data. (????)[this may be incomplete]

UPD of maternal chromosome 7 is associated with a phenotype similar to Russell-Silver syndrome with intrauterine growth restriction [36]. As shown in the pathogenic and likely pathogenic CNVs in Table 2, case 9 exhibited a de novo 11.45 Mb gene deletion at the 7q21.11q21.3 loci related to Russell-Silver syndrome; both parents had a normal chromosome 7. Actually, aCGH cannot determine the UPD condition in a patient, however, case 7 shows a loss of copy has the potential to result in an isodisomy-related phenotype. Unfortunately, the methylation analysis is not available and we lack data to rule out the disorder. However, as this imprinting disorder is characterized by pre- and postnatal growth retardation, a triangular face, and facial, limb or truncal asymmetry, the parents decided to terminate the pregnancy. Another case exhibited a maternally inherited 3.805 Mb 7q21.12q21.2 duplication classified as a VOUS. Because the risk of heterodisomy or isodisomy is very low in a case with the gain of imprinting loci, the parents decided to continue the pregnancy and a healthy baby was born (supplementary table). Neither of these cases met the criteria for UPD.

Response: Thanks for the reviewer's suggestion.

We have revised the following paragraph as suggested.

 “Most occurrences of uniparental disomy (UPD) in chromosomes do not result in phenotypic anomalies. Maternal UPD involving chromosomes 2, 7, 14, and 15 and pa-ternal UPD involving chromosomes 6, 11, 15, and 20 are associated with phenotypic growth, neurodevelopmental and behavioral abnormalities [27, 35]. UPD of maternal chromosome 7 is associated with a phenotype similar to Russell-Silver syndrome with intrauterine growth restriction [36]. As shown in the pathogenic and likely pathogenic CNVs in Table 2, case 9 exhibited a de novo 11.45 Mb gene deletion at the 7q21.11q21.3 loci related to Russell-Silver syndrome; both parents had a normal chromosome 7. Actually, aCGH cannot determine the UPD condition in a patient, however, case 7 shows a loss of copy has the potential to result in an isodisomy-related phenotype. Unfortunately, the methylation analysis is not available and we lack data to rule out the disorder. As this imprinting disorder is characterized by pre- and postnatal growth retardation, a triangular face, and facial, limb or truncal asymmetry, the parents decided to terminate the pregnancy. Another case exhibited a maternally inherited 3.805 Mb 7q21.12q21.2 du-plication classified as a VOUS. Because the risk of heterodisomy or isodisomy is very low in a case with the gain of imprinting loci, the parents decided to continue the pregnancy and a healthy baby was born (supplementary table). Neither of these cases met the criteria for UPD.”

Point 6. I will assume you have karyotyping that distinguishes the supernumerary bisatellited marker chromosome of Cat Eye (CES) with 4 copies of that region from a duplication of 22q11.2 which is 3 copies of a slightly more distal region. The phenotypes are quite different since CES can have significant cardiac and anal malformations. CES has been known to have 4 copies of the region since 1986 (PMID 3961499) also Mears 1994 (PMID 7912885). It stands out when looking at the Table that CES is listed as having 3 copies rather than the accepted 4. If the array cannot diagnose or differentiate between these two entities with distinct phenotypes, I would recommend either removing those cases or explaining why you have cases of CES with 3 copies of the region. If you have the karyotypes, that should address it. The more recent array platforms seem to do a better job with regions like this but since the platform used here targets this region, it should be accurately interpreted and diagnosed.

Response: Thanks for the reviewer's suggestion.

We have revised section 4.7 and table 2.

Thirdly, both case 20 (de novo) and case 21 (maternally inherited) involved 22q11.1q11.21 duplication affecting the morbid genes TBX1 (MIM: 602054) and CECR2 (MIM: 115470), furthermore, the duplication region is partial overlapping with a critical region of Cat Eye Syndrome (CES). CES is characterized by large phenotypic variability, ranging from near normal to severe malformations, as reflected by varied neurodevelopmental outcomes [43]. In general, CES is typically associated with a supernumerary bisatellited marker chromosome (inv dup 22pter-22q11.2) resulting in four copies of this region but aCGH cannot well discriminate the difference between 3 and 4 copies thus we further checked karyotype data of both cases. Case 20 showed mosaic karyotype with supernumerary marker chromosome (47, XY+ mar de novo[42]/46, XY,[29]) and pyelectasis was found by ultrasound. Therefore, the fetus was diagnosed as a CES, and finally the parents decided to TOP. However, the gain of CNV in case 21 was maternally inherited and karyotype was normal. Because some publications mentioned about that interstitial duplication of 22q11.2 is associated with typical CES 〔44〕. Based on those reports it cannot rule out the risk of pathogenicity and is considered as likely pathogenic case. The parents decided to continue the pregnancy and a healthy baby was born, after three years following, no significant clinical issues were found.

  1. This response is perfect, thank you.

 Three distinct neurodevelopmental disorders arise primarily from deletions or duplications that occur at the 15q11-q13 locus: Prader-Willi syndrome (PWS), Angelman syndrome (AS), and 15q11-q13 duplication syndrome (Dup15q syndrome). Each of these disorders results from the loss of function or over-expression of at least one imprinted gene. In supplementary data, we found a gain of 15q13.3 region (15q13.3 duplication) in case 12 and 16, and a loss of 15q11.2(15q11.2 deletion) in case 15, the aberrant regions are not including the major causative genes of Prader-Willi and Angelman syndrome.

 Finally, on re-review, I would like to point out:

  1. A)    For section 4.3 where the detection ability of this study and this single platform is discussed, it may be helpful for the reader if the authors also make a statement about the relative improved detection ability across the platforms that are analyzed in the Table.

Response: Thanks for the reviewer's suggestion.

We have added the below paragraph in section 4.3.

According to our review, CGH+SNP almost dominates the current prenatal testing platform. CGH+SNP can provide more clinical information on genetic findings, such as smaller CNV, UPD or tetraploidy, etc, and those findings provide valuable information to consider the possible clinical outcomes. However, for some findings, further specific tests are necessary to confirm the diagnosis, for instance, a methylation test is still necessary to rule out whether loss of imprinting is expressed in the fetus results in UPD disorder, thus it still a difficult and urgent condition in prenatal diagnosis. We recognize that the CGH platform may not provide the information as the SNP array done. In order to make up for this shortcoming, our strategy is to conduct the karyotype and aCGH side by side, the results show that targeted aCGH indeed can help discover most of significant CNVs to improve the diagnosis rate.

  1. B)    At the end of section 4.5 there is a speculative point about CMA and congenital heart defects. This is well recognized, I would recommend providing some references. For example: Landstrom et al, 2021, AHA statement; Wu et al 2017 PMD 5418813, and reviewed in Zaidi and Brueckner 2017, Circ Res

 Response: Thanks for the reviewer's suggestion.

We already add Landstrom et al, 2021, AHA statement; and Zaidi and Brueckner 2017, Circ Res into the reference

  1. C)    In section 4.6, there is a statement about increased findings in AMA patients. Please clarify: while aneuploidy (e.g. trisomies) is well known to be associated with AMA.  However, in general are CNVs (submicroscopic, particularly those involving recurrent low copy repeat mediated CNVs such as those on chr 7q, 15q, 17, 22q etc) associated with AMA??If this is the case, please provide references.If not, please modify.

Response: Thanks for the reviewer's suggestion. Modify as follow:

In this study (Table 1), AMA was the major indication for CMA, and 0.55% (3/546) of reported CNVs ≧ 400 kb lower than both literatures report. However, in the non-reported CNVs ≧ 400 kb 2.56% (14/546) were more common than reported in AMA group.

  1. D) Please review the last inserted revised section for grammar and writing flow